# Samba: Synchronized Set-of-Sequences Modeling for Multiple Object Tracking

**Mattia Segu**[1,3]**, Luigi Piccinelli**[1]**, Siyuan Li**[1]**, Yung-Hsu Yang**[1]**, Luc Van Gool**[1,2]**, Bernt Schiele**[3]

[1] ETH Zurich,   [2] INSAIT, Sofia University, St. Kliment Ohridski,
[3] Max Planck Institute for Informatics, Saarland Informatics Campus
`https://sambamotr.github.io/`

## Abstract

Multiple object tracking in complex scenarios - such as coordinated dance performances, team sports, or dynamic animal groups - presents unique challenges. In these settings, objects frequently move in coordinated patterns, occlude each other, and exhibit long-term dependencies in their trajectories. However, it remains a key open research question on how to model long-range dependencies within tracklets, interdependencies among tracklets, and the associated temporal occlusions. To this end, we introduce Samba, a novel linear-time set-of-sequences model designed to jointly process multiple tracklets by synchronizing the multiple selective state-spaces used to model each tracklet. Samba autoregressively predicts the future track query for each sequence while maintaining synchronized long-term memory representations across tracklets. By integrating Samba into a tracking-by-propagation framework, we propose SambaMOTR, the first tracker effectively addressing the aforementioned issues, including long-range dependencies, tracklet interdependencies, and temporal occlusions. Additionally, we introduce an effective technique for dealing with uncertain observations (MaskObs) and an efficient training recipe to scale SambaMOTR to longer sequences. By modeling long-range dependencies and interactions among tracked objects, SambaMOTR implicitly learns to track objects accurately through occlusions without any handcrafted heuristics. Our approach significantly surpasses prior state-of-the-art on the DanceTrack, BFT, and SportsMOT datasets.

## 1 Introduction

Multiple object tracking (MOT) involves detecting multiple objects while keeping track of individual instances throughout a video stream. It is critical for multiple downstream tasks such as sports analysis, autonomous navigation, and media production (Luo et al., 2021). Traditionally, MOT methods are validated on relatively simple settings such as surveillance datasets (Milan et al., 2016), where pedestrians exhibit largely linear motion and diverse appearance, and rarely interact with each other in complex ways. However, in dynamic environments like team sports, dance performances, or animal groups, objects frequently move in coordinated patterns, occlude each other, and exhibit non-linear motion with long-term dependencies in their trajectories (Fig. 1). Modeling the long-term interdependencies between objects in these settings, where their movements are often synchronized or influenced by one another, remains an open problem that current methods fail to address.

Current *tracking-by-detection* methods (Bewley et al., 2016; Wojke et al., 2017; Zhang et al., 2022; Cao et al., 2023) often rely on heuristics-based models like the Kalman filter to independently model the trajectory of objects and predict their future location. However, these methods struggle with the non-linear nature of object dynamics such as motion, appearance, and pose changes. *Tracking-by-propagation* (Sun et al., 2020; Meinhardt et al., 2022; Zeng et al., 2022) offers an alternative by modeling tracking as an end-to-end autoregressive object detection problem, leveraging detection transformers (Carion et al., 2020; Zhu et al., 2020) to propagate track queries over time. Their flexible design fostered promising performance in settings with complex motion, pose, and appearance patterns, such as dance (Sun et al., 2022), sports (Cui et al., 2023), and bird (Zheng et al., 2024) tracking datasets. However, such methods only propagate the temporal information across adjacent frames, failing to account for long-range dependencies. MeMOTR (Gao & Wang, 2023) attempts a preliminary solution to this problem by storing temporal information through an external heuristics-

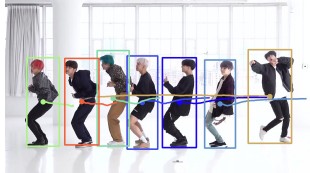 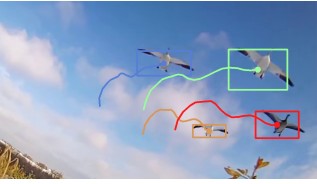 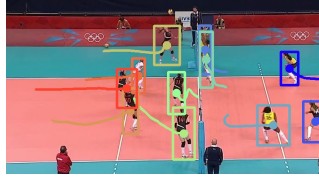

(a) DanceTrack (Sun et al., 2022)    (b) BFT (Zheng et al., 2024)    (c) SportsMOT (Cui et al., 2023)

Figure 1: **Tracking multiple objects in challenging scenarios** - such as coordinated dance performances (a), dynamic animal groups (b), and team sports (c) - requires handling complex interactions, occlusions, and fast movements. As shown in the tracklets above, objects may move in coordinated patterns and occlude each other. By leveraging the joint long-range dependencies in their trajectories, SambaMOTR accurately tracks objects through time and occlusions.

based memory. However, its use of an exponential moving average (EMA) to compress past history results in a suboptimal temporal memory representation, as it discards fine-grained long-range dependencies that are crucial for accurate tracking over time. Moreover, by processing each tracklet independently and overlooking tracklets interaction, current methods cannot accurately model objects' behavior through occlusions, resorting to naive heuristics to handle such cases: some (Zhang et al., 2023; Gao & Wang, 2023) freeze the track queries during occlusions and only rely on their last observed state during prolonged occlusions; others (Zeng et al., 2022) delegate occlusion management to the propagation module, which fails to estimate accurate track trajectories as it only propagates information across adjacent frames and does not account for historical information. We argue that effective long-term memory and interaction modeling allow for more accurate inference of occluded objects' behavior in complex environments, such as team sports or dance performances, by leveraging past information and understanding joint motion patterns.

To address these shortcomings, we propose Samba [1] , a novel linear-time set-of-sequences model that processes a set of sequences (*e.g.* multiple tracklets) simultaneously and compresses their histories into synchronized long-term memory representations, capturing interdependencies within the set. Samba adopts selective SSMs from Mamba (Gu & Dao, 2023) to independently model all tracklets, compressing their long-range histories into hidden states. We then propose to synchronize these memory representations across tracklets at each time step to account for interdependencies (*e.g.* interactions among tracklets). We implement synchronization via a self-attention mechanism (Vaswani et al., 2017) across the hidden states of all sequences, allowing tracklets to exchange information. This approach proves beneficial in datasets where objects move in coordinated patterns (Tabs. 1 to 3). The resulting set-of-sequences model, Samba, retains the linear-time complexity of SSMs while modeling the joint dynamics of the set of tracklets.

By integrating Samba into a tracking-by-propagation framework (Zeng et al., 2022), we present SambaMOTR, an end-to-end multiple object tracker that models long-range dependencies and interactions between tracklets to handle complex motion patterns and occlusions in a principled manner. SambaMOTR complements a transformer-based object detector with a novel set-of-queries propagation module based on Samba, which accounts for both individual tracklet histories and their interactions when autoregressively predicting the next track queries.

Additionally, some queries result in uncertain detections due to occlusions or challenging scenarios (see Fig. 2, Occlusion). To prevent these detections from compromising the memory representation and accumulating errors during query propagation with Samba, we propose MaskObs. MaskObs blocks unreliable observations from entering the set-of-queries propagation module while updating the corresponding hidden states and track queries using only the long-term memory of their tracklets and interactions with confidently tracked objects. Unlike previous methods that freeze track queries during occlusions, MaskObs leverages both temporal and spatial context - *i.e.* past behavior and interdependencies with other tracklets - to more accurately predict an object's future state. Consequently, SambaMOTR tracks objects through occlusions more effectively (Tab. 4, line d).

---

[1]Samba is named for its foundation on the **S**ynchronization of **Mamba**'s selective state-spaces (Gu & Dao, 2023). Its name reflects the coordinated motion of tracklets (Fig. 1), much like the synchronized movements in the samba dance. By synchronizing the hidden states across sequences, our approach is disentangled from Mamba's selective state-space models (SSMs) and can, in principle, be applied to any sequence model that includes an intermediate memory representation, such as other SSMs or recurrent neural networks.

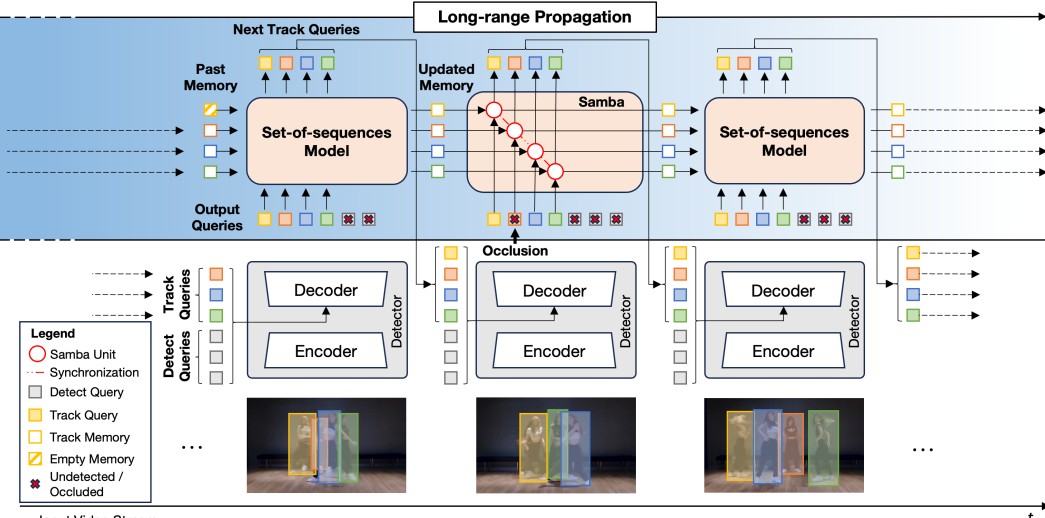

Figure 2: **Overview of SambaMOTR.** SambaMOTR combines a transformer-based object detector with a set-of-sequences Samba model. The object detector's encoder extracts image features from each frame, which are fed into its decoder together with detect and track queries to detect newborn objects or re-detect tracked ones. The Samba set-of-sequences model is composed of multiple synchronized Samba units that simultaneously process the past memory and currently observed output queries for all tracklets to predict the next track queries and update the track memory. The hidden states of newborn objects are initialized from zero values (barred squares). In case of occlusions or uncertain detections, the corresponding query is masked (red cross) during the Samba update.

Finally, we introduce an efficient training recipe to scale SambaMOTR to longer sequences by sampling arbitrarily long sequences, computing tracking results, and applying gradients only on the last five frames. This simple strategy enables us to learn longer-range dependencies for query propagation, improving the tracking performance while maintaining the same GPU memory requirements as previous methods (Zeng et al., 2022; Gao & Wang, 2023).

We validate SambaMOTR on the challenging DanceTrack (Sun et al., 2022), SportsMOT (Cui et al., 2023), and BFT (Zheng et al., 2024) datasets. Owing to our contributions, we establish a new state of the art on all datasets. We summarize them as follows: **(a)** we introduce Samba, our novel linear-time set-of-sequences model based on synchronized SSMs; **(b)** we introduce SambaMOTR, the first tracking-by-propagation method that leverages past tracklet history in a principled manner to learn long-range dependencies, tracklets interaction, and occlusion handling; **(c)** we introduce MaskObs, a simple technique for dealing with uncertain observations in SSMs and an efficient training recipe that enables learning stronger sequence models with limited compute.

## 2 RELATED WORK

**Tracking-by-detection.** *Tracking-by-detection* is a popular paradigm in MOT, consisting of an object detection stage followed by data association to yield object trajectories throughout a video. Motion and appearance cues are typically utilized to match detections to tracklets through hand-crafted heuristics. The motion-based tracker SORT (Bewley et al., 2016) relies on Intersection over Union (IoU) to assign the tracklet locations predicted with a Kalman filter to object detections. Byte-Track (Zhang et al., 2022) introduces a two-stage matching scheme to associate low-confidence detections. OC-SORT (Cao et al., 2023) models non-linear motion by taking care of noise accumulation under occlusion. Alternatively, appearance descriptors can be used alone (Pang et al., 2021; Li et al., 2022; 2024a) or in combination with motion (Wojke et al., 2017; Wang et al., 2020; Zhang et al., 2021; Segu et al., 2024; Li et al., 2024b) to match detections to tracklets according to a similarity metric. Due to the disentangled nature of the two stages, tracking-by-detection methods historically leveraged state-of-the-art object detectors to top the MOT challenge (Dendorfer et al., 2021). However, by relying on hand-crafted heuristics, such methods struggle with non-linear motion and appearance patterns (Sun et al., 2022; Cui et al., 2023; Zheng et al., 2024; Li et al., 2023), and require domain-specific hyperparameters (Segu et al., 2023; Liu et al., 2023). Recent transformer-

based methods have eased the burden of heuristics. TransTrack (Sun et al., 2020) decodes track and detect queries with siamese transformer decoders and associates them with simple IoU matching. MeMOT (Cai et al., 2022) fuses a large memory bank into a tracklet descriptor with a transformer-based memory aggregator. However, it requires storing and processing with quadratic complexity the historical information from up to 27 past frames. GTR (Zhou et al., 2022) matches static trajectory queries to detections to generate tracklets, but fails to model object motion and tracklet interaction. In contrast, SambaMOTR implicitly learns motion, appearance, and tracklet interaction models by autoregressively predicting the future track queries with our set-of-sequences model Samba.

**Tracking-by-propagation.** Recent work (Meinhardt et al., 2022; Zeng et al., 2022) introduced a more flexible and end-to-end trainable *tracking-by-propagation* design that treats MOT as an autoregressive problem where object detection and query propagation are tightly coupled. Leveraging the transformer-based Deformable DETR (Zhu et al., 2020) object detector, TrackFormer (Meinhardt et al., 2022) and MOTR (Zeng et al., 2022) autoregressively propagate the detection queries through time to re-detect (track) the same object in subsequent frames. MOTRv2 (Zhang et al., 2023) leverages a pre-trained YOLOX object detector to provide anchors for Deformable DETR and boost its detection performance. However, these approaches only propagate queries across adjacent frames, failing to fully leverage the historical information. MeMOTR (Gao & Wang, 2023) first attempts to utilize the temporal information in tracking-by-propagation by aggregating long- (EMA of a tracklet's queries through time) and short-term memory (fusion of the output detect queries across the last two observed frames) in a temporal interaction module. By collapsing the tracklet history with an EMA and by freezing the last observed state of track queries and memory under occlusions, MeMOTR cannot accurately estimate track query trajectories through occlusions. Finally, by modeling query propagation independently for each tracklet, it does not model tracklet interaction. In contrast, our proposed Samba set-of-sequences model relies on individual SSMs to independently model each tracklet as a sequence and it synchronizes the memory representations across the set of tracklets to enable tracklet interaction. Equipped with Samba, SambaMOTR autoregressively predicts future queries aware of long-range dynamics and of other tracklets' motion and appearance.

## 3 PRELIMINARIES

Before introducing SambaMOTR (Sec. 4), we present the necessary background and notation on selective state-space models (Sec. 3.1) and tracking-by-propagation (Sec. 3.2).

### 3.1 SELECTIVE STATE-SPACE MODELS

Inspired by classical state-space models (SSMs) (Kalman, 1960), structured SSMs (S4) (Gu et al., 2021) introduce a sequence model whose computational complexity scales linearly, rather than quadratically, with the sequence length. This makes S4 a principled and efficient alternative to transformers (Vaswani et al., 2017). By further introducing a selection mechanism - *i.e.* rendering the SSM parameters input-dependent - Mamba (Gu & Dao, 2023) can model time-variant systems, bridging the performance gap with transformers (Vaswani et al., 2017).

We here formally define selective SSMs (S6) (Gu & Dao, 2023). Let $x(t)$ be the input signal at time $t$, $h(t)$ the hidden state, and $y(t)$ the output signal. Given the system $\mathbf{A}$, control $\mathbf{B}$, and output $\mathbf{C}$ matrices, we define the continuous linear time-variant SSM in Eq. (1). The discrete-time equivalent system (Eq. (2)) of the defined SSM is obtained through a discretization rule. The chosen discretization rule is typically the zero-order hold (ZOH) model: $\bar{\mathbf{A}}(t) = \exp(\mathbf{\Delta}(t)\mathbf{A}(t))$, $\bar{\mathbf{B}}(t) = (\mathbf{\Delta}(t)\mathbf{A}(t))^{-1}(\exp(\mathbf{\Delta}(t)\mathbf{A}(t)) - \mathbf{I}) \cdot \mathbf{\Delta}(t)\mathbf{B}(t)$:

$$h'(t) = \mathbf{A}(t)h(t) + \mathbf{B}(t)x(t) \qquad (1) \qquad\qquad h_t = \bar{\mathbf{A}}(t)h_{t-1} + \bar{\mathbf{B}}(t)x_t \qquad (2)$$
$$y(t) = \mathbf{C}(t)h(t) \qquad\qquad\qquad\qquad y_t = \mathbf{C}(t)h_t$$

$x_t$, $h_t$, $y_t$ are the observations sampled at time $t$ of the input signal $x(t)$, hidden state $h(t)$, and output signal $y(t)$. While S4 learns a linear time-invariant (LTI) system with $\mathbf{\Delta}(t) = \mathbf{\Delta}$, $\mathbf{A}(t) = \mathbf{A}$, $\mathbf{B}(t) = \mathbf{B}$ and $\mathbf{C}(t) = \mathbf{C}$, S6 introduces selectivity to learn a time-variant system by making $\mathbf{\Delta}(t)$, $\mathbf{B}(t)$ and $\mathbf{C}(t)$ dependent on the input $x(t)$, *i.e.* $\mathbf{\Delta}(t) = \tau_\Delta(\mathbf{\Delta} + s_\Delta(x(t)))$, $\mathbf{B}(t) = s_B(x(t))$, $\mathbf{C}(t) = s_C(x(t))$, where $\tau_\Delta = \texttt{softplus}$, and $s_\Delta$, $s_B$, $s_C$ are learnable linear mappings.

In this paper, we propose to treat tracking-by-propagation as a sequence modeling problem. Given the discrete sequence of historical track queries for a certain tracklet, our query propagation mod-

ule Samba (Sec. 4.2) leverages SSMs to account for the historical tracklet information in a principled manner. By recursively compressing all tracklet history into a long-term memory, Samba's complexity scales linearly with the number of frames, enabling efficient training on long sequences while processing indefinitely long tracklets at inference time.

## 3.2 Tracking-by-propagation

Tracking-by-propagation methods alternate between a detection stage and a propagation stage, relying on a DETR-like (Carion et al., 2020) transformer object detector and a query propagation module. At a time step $t$, the backbone and transformer encoder extract image features for a frame $I_t$. The *detection stage* involves feeding both a fixed-length set of learnable detect queries $Q_t^{det}$ to the transformer decoder to detect newborn objects and a variable-length set of propagated track queries $Q_t^{tck}$ to re-detect tracked ones. At time $t = 0$, the set of track queries is empty, *i.e.* $Q_0^{tck} = E_0^{tck} = \emptyset$. Detect and track queries $[Q_t^{det}, Q_t^{tck}]$ interact in the decoder with image features to generate the corresponding output embeddings $[E_t^{det}, E_t^{tck}]$ and bounding box predictions $[D_t^{det}, D_t^{tck}]$. We denote the set of embeddings corresponding to newborn objects as $\hat{E}_t^{det}$, and $\hat{E}_t^{tck} = [\hat{E}_t^{det}, E_t^{tck}]$ as the set of embeddings corresponding to the tracklets $\mathcal{S}_t$ active at time $t$. During the *propagation stage*, a query propagation module $\Theta(\cdot)$ typically takes as input the set of embeddings $\hat{E}_t^{tck}$ and outputs refined tracked queries $Q_{t+1}^{tck} = \Theta(\hat{E}_t^{tck})$ to re-detect the corresponding objects in the next frame.

Although prior work failed to properly model long-range history and tracklet interactions (Zeng et al., 2022; Gao & Wang, 2023; Meinhardt et al., 2022), and given that multiple objects often move synchronously (Fig. 1), we argue that the future state of objects in a scene can be better predicted by (i) considering both their historical positions and appearances, and (ii) estimating their interactions. In this work, we cast query propagation as a set-of-sequences modeling problem. Given a set of multiple tracklets, we encode the history of each tracklet in a memory representation using a state-space model and propose memory synchronization to account for their joint dynamics.

## 4 Method

In this section, we introduce SambaMOTR, an end-to-end multiple object tracker that combines transformer-based object detection with our set-of-sequences model Samba to jointly model the long-range history of each tracklet and the interaction across tracklets to propagate queries. First, in Sec. 3.2 we provide background on the tracking-by-propagation framework and motivate the need for better modeling of both temporal information and tracklets interaction. Then, we describe the SambaMOTR architecture (Sec. 4.1) and introduce Samba (Sec. 4.2), our novel set-of-sequences model based on synchronized state spaces that jointly models the temporal dynamics of a set of sequences and their interdependencies. Finally, in Sec. 4.3 we describe SambaMOTR's query propagation strategy based on Samba, our effective technique MaskObs to deal with occlusions in SSMs, a recipe to learn long-range sequence models with limited compute, and our simple inference pipeline.

### 4.1 Architecture

Similar to other tracking-by-propagation methods (Meinhardt et al., 2022; Zeng et al., 2022; Gao & Wang, 2023), the proposed SambaMOTR architecture (Fig. 2) is composed of a DETR-like (Carion et al., 2020) object detector and a query propagation module. As object detector, we use Deformable-DETR (Zhu et al., 2020) with a ResNet-50 (He et al., 2016) backbone followed by a transformer encoder to extract image features and a transformer decoder to detect bounding boxes from a set of detect and track queries. As query propagation module, we use our set-of-sequences model Samba. Each sequence is processed by a Samba unit synchronized with all others. A Samba unit consists of two Samba blocks (Sec. 4.2) interleaved with LayerNorm (Ba et al., 2016) and a residual connection.

### 4.2 Samba: Synchronized State-Space Models for Set-of-sequences Modeling

Set-of-sequences modeling involves simultaneously modeling a set of temporal sequences and the interdependencies among them. In MOT, set-of-sequences models can capture long-range temporal relationships within each tracklet as well as complex interactions across tracklets. To this end, we introduce Samba, a linear-time set-of-sequences model based on the synchronization of multiple state-space models. In this paper, we leverage Samba as a set-of-queries propagation network to jointly model multiple tracklets and their interactions in a tracking-by-propagation framework.

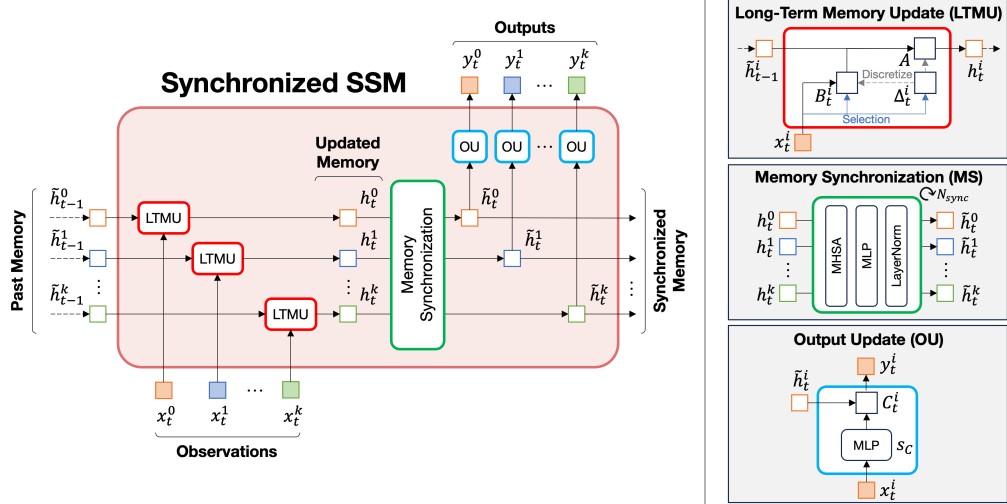

Figure 3: **Synchronized State-Space Models.** We illustrate a set of $k$ synchronized SSMs. A Long-Term Memory Update block updates each hidden state $\tilde{h}_{t-1}^i$ based on the current observation $x_t^i$, resulting in the updated memory $h_t^i$. The Memory Synchronization block then derives the synchronized hidden state $\tilde{h}_t^i$, which is fed into the Output Update module to predict the output $y_t^i$.

**Synchronized Selective State-Space Models.** Let $x_t^i$ be the discrete observation at time $t$ of the $i$-th input sequence from a set of sequences $\mathcal{S}$. We choose selective SSMs (Gu & Dao, 2023) to model each sequence through a hidden state $h_t^i$ (Eq. (3a)) encoding long-term memory, but our approach applies to any other SSM. Given the memory $\tilde{h}_{t-1}^i$, we define a long-term memory update (LTMU) function that updates $\tilde{h}_{t-1}^i$ based on the current observation $x_t^i$, resulting in the updated memory $h_t^i$. We propose a memory synchronization (MS) function $\Gamma_{i\in\mathcal{S}}(\cdot)$ that produces a set of synchronized hidden states $\tilde{h}_t^i \ \forall i \in \mathcal{S}$, modeling interactions across the set of sequences $\mathcal{S}$ (Eq. (3b)). Finally, we derive the output $y_t^i$ for each sequence through the output update (OU) function (Eq. (3c)).

$$\text{LTMU:} \qquad h_t^i = \bar{\mathbf{A}}^i(t)h_{t-1}^i + \bar{\mathbf{B}}^i(t)x_t^i \tag{3a}$$

$$\text{MS:} \qquad \tilde{h}_t^i = \Gamma_{i\in\mathcal{S}}\left(h_t^i\right) \tag{3b}$$

$$\text{OU:} \qquad y_t^i = \mathbf{C}(t)\tilde{h}_t^i \tag{3c}$$

An ideal memory synchronization function should be flexible regarding the number of its inputs (hidden states) and equivariant to their order. Thus, we propose to define the memory synchronization function $\Gamma(\cdot) = [FFN(MHSA(\cdot))]_{\times N_{sync}}$ as a set of $N_{sync}$ stacked blocks with multi-head self-attention (MHSA) (Vaswani et al., 2017) followed by a feed-forward network (FFN). A schematic illustration of the proposed synchronized state-space model layer is in Fig. 3.

**Set-of-sequences Model.** Refer to App. B.1 for a detailed description of how our synchronized SSM is used in the Samba units that model each sequence in the set-of-sequences Samba model.

### 4.3 SAMBAMOTR: END-TO-END TRACKING-BY-PROPAGATION WITH SAMBA

**Query Propagation with Samba.** As described in Sec. 3.2, the query propagation module $\Theta(\cdot)$ takes as input the decoder output embeddings $\hat{E}_t^{tck}$ and outputs refined track queries $Q_{t+1}^{tck} = \Theta(\hat{E}_t^{tck})$. SambaMOTR extends this paradigm by accounting for the temporal information and tracklets interaction. In particular, we use our Samba module $\Phi(\cdot)$ to compress the history of each tracklet into a hidden state $h_t^i$ and synchronize it across tracklets to derive the synchronized memory $\tilde{h}_t^i$. Notice that $\tilde{h}_t^i = \mathbf{0}$ for a newborn object $i$. At time $t$, we first enrich the detector output embeddings $\hat{E}_t^{tck}$ with position information by summing to them sine-cosine positional encodings $PE(\cdot)$ of the corresponding bounding boxes coordinates $\hat{D}_t^{tck}$ to implicitly model object motion and appearance, obtaining the set of input observations $X_t^{tck} = \hat{E}_t^{tck} + PE(\hat{D}_t^{tck})$. Given the set of input observations $X_t^{tck}$ and past synchronized hidden states $\tilde{H}_{t-1}$ for all tracklets in the set

$\mathcal{S}_t$ of tracklets at time $t$, we feed them into Samba $(Y_t, \tilde{H}_t) = \Phi(X_t^{tck}, \tilde{H}_{t-1})$ to obtain the output embeddings $Y_t$ and updated synchronized hidden states $\tilde{H}_t$. Finally, we use the output embeddings $Y_t$ and a learnable mapping $s_y$ to predict a residual $\Delta Q_t^{tck} = s_y(Y_t)$ to the past track queries $Q_t^{tck}$ and generate the new ones, *i.e.* $Q_{t+1}^{tck} = Q_t^{tck} + \Delta Q_t^{tck}$. By recursively unfolding this process over time, SambaMOTR can track multiple objects while compressing indefinitely long tracklet histories into their long-term memory representations, effectively modeling object motion and appearance changes, and tracklets interactions.

**MaskObs: Dealing with Uncertain Observations.** Tracking-by-propagation may occasionally deal with occluded objects or uncertain detections. Given a function $conf(\cdot)$ to estimate the predictive confidence of an input observation $x_t^i$, we propose MaskObs, a strategy to handle uncertain observations. MaskObs masks uncertain observations from the state update (Eq. (4)), thus defining the system dynamics solely based on its history and the interdependencies with other sequences:

$$h_t^i = \bar{\mathbf{A}}^{\mathbf{i}}(t)h_{t-1}^i + \bar{\mathbf{B}}^{\mathbf{i}}(t)x_t^i \cdot \mathbb{1}[conf(x_t^i) > \tau_{mask}] \tag{4}$$

$\mathbb{1}[\cdot]$ is the indicator function, and $\tau_{mask}$ is the confidence threshold, *e.g.* $\tau_{mask} = 0.5$. We implement $conf(x_t^i)$ as the predictive confidence $conf(d_t^i)$ of the corresponding bounding box $d_t^i$. In our work, this design choice allows us to better model query propagation through occlusions (Tab. 4, line b).

**Efficiently Learning Long-range Sequence models.** Previous MOTR-like approaches are trained end-to-end on a sequence of 5 consecutive frames sampled at random intervals. While SambaMOTR's set-of-sequences model Samba already shows impressive generalization performance to long sequences at inference time (Tab. 4, line c), we propose to train on longer sequences (*i.e.* 10 frames) and only apply gradients to the last 5 frames (Tab. 4, line d). We hypothesize that this strategy allows us to learn better history compression for late observations in a sequence, resulting in even better tracking performance while being trained with similar GPU memory requirements. A schematic illustration of our training scheme proposal is in Fig. B.

**Inference Pipeline.** At a given time step $t$, we jointly input the learnable detect queries $Q_{det}$ and track queries $Q_t^{tck}$ ($Q_0^{tck} = \emptyset$) into the transformer decoder to produce detection embeddings $E_t^{det}$ and tracking embeddings $E_t^{tck}$ and the corresponding bounding boxes. Each detection bounding box with a confidence score higher than a threshold $\tau_{det}$ will initialize a newborn track $\hat{E}^{det}$. We then propagate the embeddings of newborn $\hat{E}^{det}$ and tracked $E_t^{tck}$ objects together with the track memory $\tilde{H}_{t-1}$ to generate the updated track queries $Q_{track}^{t+1}$ and synchronized memory $\tilde{H}_{t+1}$. To deal with occlusions and lost objects, we consider an individual track query $q_t^{i,track}$ inactive if its corresponding bounding box confidence $conf(d_t^i)$ at time $t$ is lower than $\tau_{track}$. If a track query is inactive for more than $N_{miss}$ frames, it is deemed lost and dropped.

Unlike MeMOTR (Gao & Wang, 2023), which does not update the track embedding and long-term memory for an object with low detection confidence at a time step $t$, our approach employs a principled query propagation scheme that can hallucinate likely track query trajectories under occlusions by relying on its past history or attending to other trajectories. Thus, we always update the memory and track query for any tracklet - even when occluded - as long as it is not deemed lost.

## 5 EXPERIMENTS

In this section, we present experimental results to validate SambaMOTR. We describe our evaluation protocol (Sec. 5.1) and report implementation details (Sec. 5.2). We then compare SambaMOTR to the previous state-of-the-art methods (Sec. 5.3) and conduct an ablation study (Sec. 5.4) on the method components. We provide more ablations in the appendix. Qualitative results can be found in Fig. 1 and at the anonymous project page `https://anonymous-samba.github.io/`.

### 5.1 EVALUATION PROTOCOL

**Datasets.** To evaluate SambaMOTR, we select a variety of challenging datasets exhibiting highly non-linear motion in crowded scenarios, with frequent occlusions and uniform appearances. All datasets present scenes with objects moving synchronously. Thus, they represent a suitable benchmark for assessing the importance of modeling tracklet interaction. DanceTrack (Sun et al., 2022) is a multi-human tracking dataset composed of 100 group dancing videos. The Bird Flock Tracking (BFT) (Zheng et al., 2024) dataset includes 106 clips from the BBC documentary series Earthflight (Downer & Tennant, 2011). SportsMOT (Cui et al., 2023) consists of 240 video sequences

from basketball, volleyball, and soccer scenes. Due to the highly linear motion in MOT17 (Milan et al., 2016), its small size (only 7 videos), and the subsequent need for training on additional detection datasets, end-to-end tracking methods do not provide additional advantages over more naive Kalman-filter-based methods. We report its results in the Appendix.

**Metrics.** Following prior work, we measure the overall tracking performance with the HOTA (Luiten et al., 2021) metric and disentangle detection accuracy (DetA) and association accuracy (AssA). We report the MOTA (Bernardin & Stiefelhagen, 2008) and IDF1 (Ristani et al., 2016) metrics for completeness. Since our objective is improving association performance and the overall tracking quality, HOTA and AssA are the most representative metrics.

## 5.2 Implementation Details

Following prior works (Gao & Wang, 2023; Zhang et al., 2023), we apply random resize, random crop, and photometric augmentations as data augmentation. The shorter side of the input image is resized to 800 preserving the aspect ratio, and the maximum size is restricted to 1536. For a fair comparison with prior work (Sun et al., 2020; Zeng et al., 2022; Gao & Wang, 2023), we use the Deformable-DETR (Zhu et al., 2020) object detector with ResNet-50 (He et al., 2016) and initialize it from COCO (Lin et al., 2014) pre-trained weights. Similar to MeMOTR (Gao & Wang, 2023), we inject track queries after one decoder layer. We run our experiments on 8 NVIDIA RTX 4090 GPUs, with batch size 1 per GPU. Each batch element contains a video clip with 10 frames, and we compute and backpropagate the gradients only over the last 5. We sample uniformly spaced frames at random intervals from 1 to 10 within each clip. We utilize the AdamW optimizer (Loshchilov & Hutter, 2017) with initial learning rate of $2.0 \times 10^{-4}$. For simplicity, $\tau_{det} = \tau_{track} = \tau_{mask} = 0.5$. $N_{miss}$ is 35, 20, and 50 on DanceTrack, BFT, and SportsMOT, respectively, due to different dataset dynamics. On DanceTrack (Sun et al., 2022), we train SambaMOTR for 15 epochs on the training set and drop the learning rate by a factor of 10 at the $10^{th}$ epoch. On BFT (Sun et al., 2022), we train for 20 epochs and drop the learning rate after 10 epochs. On SportsMOT (Cui et al., 2023), we train for 18 epochs and drop the learning rate after 8 and 12 epochs. SambaMOTR's inference runs at 16 FPS on a single NVIDIA RTX 4090 GPUs.

## 5.3 Comparison with the State of the Art

We compare SambaMOTR with multiple tracking-by-detection and tracking-by-propagation approaches on the DanceTrack (Tab. 1), BFT (Tab. 2) and SportsMOT (Tab. 3) datasets. All methods are trained without using additional datasets. Since trackers use various object detectors with different baseline performance, we report the detector used for each method. For fair comparison, we report the performance of tracking-by-propagation methods with Deformable DETR (Zhu et al., 2020), marking the best in **bold**. We underle the overall best result. Tracking-by-detection methods often use the stronger YOLOX-X (Ge et al., 2021), but tracking-by-propagation consistently outperforms them, with SambaMOTR achieving the highest HOTA and AssA across all datasets.

**DanceTrack.** The combination of highly irregular motion and crowded scenes with frequent occlusions and uniform appearance historically made DanceTrack challenging for tracking-by-detection methods. Despite their higher DetA when using the strong object detector YOLOX-X (Ge et al., 2021), tracking-by-propagation significantly outperforms them (see MeMOTR (Gao & Wang, 2023) and SambaMOTR). SambaMOTR sets a new state of the art, with $+3.8$ HOTA and $+5.2$ AssA on the strongest competitor MeMOTR. Our method owes this performance improvement to its better modeling of the historical information, our effective strategy to learn accurate sequence models through occlusions, and our modeling of tracklets interaction (group dancers move synchronously).

**BFT.** Bird flocks present similar appearance and non-linear motion. For this reason, OC-SORT works best among tracking-by-detection methods. Nevertheless, bird flocks move synchronously, and interaction among tracklets is an essential cue for modeling joint object motion. Thanks to our proposed sequence models synchronization, SambaMOTR achieves $+2.8$ HOTA and $+4.9$ AssA over the best competitor overall (OC-SORT), and an impressive $+6.3$ HOTA and $+12.5$ improvement over the previous best tracking-by-propagation method TrackFormer (Meinhardt et al., 2022).

**SportsMOT.** Sports scenes typically present non-linear motion patterns that the Kalman filter struggles to model, hence the underwhelming performance of ByteTrack (Zhang et al., 2022). For this reason, trackers that model non-linear motion either explicitly (OC-SORT (Cao et al., 2023)) or im-

Table 1: **State-of-the-art comparison on DanceTrack** (Sun et al., 2022) without additional training data. Best tracking-by-propagation method in **bold**; best overall underlined.

| Methods | Detector | HOTA | AssA | DetA | IDF1 | MOTA |
|---|---|---|---|---|---|---|
| *Tracking-by-detection:* | | | | | | |
| FairMOT (Zhang et al., 2021) | | 39.7 | 23.8 | 66.7 | 40.8 | 82.2 |
| CenterTrack (Zhou et al., 2020) | CenterNet (Duan et al., 2019) | 41.8 | 22.6 | 78.1 | 35.7 | 86.8 |
| TraDeS (Wu et al., 2021) | | 43.3 | 25.4 | 74.5 | 41.2 | 86.2 |
| TransTrack (Sun et al., 2020) | Deformable DETR (Zhu et al., 2020) | 45.5 | 27.5 | 75.9 | 45.2 | 88.4 |
| GTR (Zhou et al., 2022) | CenterNet2 (Zhou et al., 2021) | 48.0 | 31.9 | 72.5 | 50.3 | 84.7 |
| ByteTrack (Zhang et al., 2022) | | 47.7 | 32.1 | 71.0 | 53.9 | 89.6 |
| QDTrack (Pang et al., 2021) | YOLOX-X (Ge et al., 2021) | 54.2 | 36.8 | 80.1 | 50.4 | 87.7 |
| OC-SORT (Cao et al., 2023) | | 55.1 | 38.3 | 80.3 | 54.6 | 92.0 |
| C-BIoU (Yang et al., 2023) | | 60.6 | 45.4 | 81.3 | 61.6 | 91.6 |
| *Tracking-by-propagation:* | | | | | | |
| MOTR (Zeng et al., 2022) | | 54.2 | 40.2 | 73.5 | 51.5 | 79.7 |
| MeMOTR (Gao & Wang, 2023) | Deformable DETR (Zhu et al., 2020) | 63.4 | 52.3 | 77.0 | 65.5 | 85.4 |
| SambaMOTR (ours) | | **67.2** | **57.5** | **78.8** | **70.5** | **88.1** |

Table 2: **State-of-the-art comparison on BFT** (Zheng et al., 2024) without additional training data. Best tracking-by-propagation method in **bold**; best overall underlined.

| Method | Detector | HOTA | AssA | DetA | IDF1 | MOTA |
|---|---|---|---|---|---|---|
| *Tracking-by-detection:* | | | | | | |
| FairMOT (Zhang et al., 2021) | CenterNet (Duan et al., 2019) | 40.2 | 28.2 | 53.3 | 41.8 | 56.0 |
| CenterTrack (Zhou et al., 2020) | | 65.0 | 54.0 | 58.5 | 61.0 | 60.2 |
| SORT (Wojke et al., 2017) | | 61.2 | 62.3 | 60.6 | 77.2 | 75.5 |
| ByteTrack (Zhang et al., 2022) | YOLOX-X (Ge et al., 2021) | 62.5 | 64.1 | 61.2 | 82.3 | 77.2 |
| OC-SORT (Cao et al., 2023) | | 66.8 | 68.7 | 65.4 | 79.3 | 77.1 |
| TransCenter (Xu et al., 2022) | Deformable DETR (Zhu et al., 2020) | 60.0 | 61.1 | 66.0 | 72.4 | 74.1 |
| TransTrack (Sun et al., 2020) | | 62.1 | 60.3 | 64.2 | 71.4 | 71.4 |
| *Tracking-by-propagation:* | | | | | | |
| TrackFormer (Meinhardt et al., 2022) | Deformable DETR (Zhu et al., 2020) | 63.3 | 61.1 | **66.0** | 72.4 | **74.1** |
| SambaMOTR (ours) | | **69.6** | **73.6** | **66.0** | **81.9** | 72.0 |

plicitly (TransTrack (Sun et al., 2020)) perform well. Notably, our tracking-by-propagation SambaMOTR enables implicit joint modeling of motion, appearance, and tracklet interaction, obtaining the best HOTA overall (69.8) despite the lower DetA of our Deformable-DETR detector compared to OC-SORT's YOLOX-X. Moreover, SambaMOTR exhibits a significant +1.6 AssA over the best tracking-by-propagation method and an impressive +4.6 AssA over OC-SORT.

## 5.4 ABLATION STUDIES

We ablate the effect of each component of our method in Tab. 4, as detailed in Sec. 4 and illustrated in Fig. B. Additional ablation studies are presented in App. C.2.

**SSM.** Line (a) shows the benefits of a sequential representation for tracking. We use a vanilla sequence model, such as Mamba, as the baseline for query propagation, establishing a robust foundation that outperforms MeMOTR's EMA-based history and temporal attention module.

**MaskObs.** Handling track queries during occlusions (line b) with MaskObs - which masks uncertain observations from the state update and relies on long-term memory and interactions with visible tracklets - leads to significant overall improvements (+1.3 HOTA), highlighting the effectiveness of managing occluded objects.

**Sync.** Making tracklets aware of each other through our synchronization mechanism (line c) results in over 1% improvement across all metrics, demonstrating how modeling interactions between tracklets enhances tracking accuracy by capturing joint dynamics and coordinated movements.

Table 3: **State-of-the-art comparison on SportsMOT** (Cui et al., 2023) without additional training data. Best tracking-by-propagation method in **bold**; best overall underlined.

| Methods | Detector | HOTA | AssA | DetA | IDF1 | MOTA |
|---|---|---|---|---|---|---|
| ***Tracking-by-detection:*** | | | | | | |
| FairMOT (Zhang et al., 2021) | CenterNet (Duan et al., 2019) | 49.3 | 34.7 | 70.2 | 53.5 | 86.4 |
| QDTrack (Pang et al., 2021) | | 60.4 | 47.2 | 77.5 | 62.3 | 90.1 |
| ByteTrack (Zhang et al., 2022) | YOLOX-X (Ge et al., 2021) | 62.1 | 50.5 | 76.5 | 69.1 | 93.4 |
| OC-SORT (Cao et al., 2023) | | 68.1 | 54.8 | 84.8 | 68.0 | 93.4 |
| TransTrack (Sun et al., 2020) | Deformable DETR (Zhu et al., 2020) | 68.9 | 57.5 | 82.7 | 71.5 | 92.6 |
| ***Tracking-by-propagation:*** | | | | | | |
| MeMOTR (Gao & Wang, 2023) | Deformable DETR (Zhu et al., 2020) | 68.8 | 57.8 | 82.0 | 69.9 | 90.2 |
| SambaMOTR (ours) | | **69.8** | **59.4** | **82.2** | **71.9** | **90.3** |

Table 4: **Ablation on method components** on the DanceTrack test set. Compared to prior work (in gray), we introduce a long-range query propagation module based on state-space models (SSM), we mask uncertain queries during the state update (MaskObs), we synchronize memory representations across tracklets (Sync), and we learn from longer sequences (Longer).

| Method | | SSM | MaskObs | Sync | Longer | HOTA | AssA | DetA | IDF1 | MOTA |
|---|---|---|---|---|---|---|---|---|---|---|
| | (a) | ✓ | - | - | - | 63.5 | 53.8 | 75.1 | 67.0 | 81.7 |
| SambaMOTR (**Ours**) | (b) | ✓ | ✓ | - | - | 64.8 | 54.3 | 77.7 | 68.1 | 85.7 |
| | (c) | ✓ | ✓ | ✓ | - | 65.9 | 55.6 | 78.4 | 68.7 | 87.4 |
| | (d) | ✓ | ✓ | ✓ | ✓ | **67.2** | **57.5** | **78.8** | **70.5** | **88.1** |
| MOTR (Zeng et al., 2022) | (e) | - | - | - | - | 54.2 | 40.2 | 73.5 | 51.5 | 79.7 |
| MeMOTR (Gao & Wang, 2023) | (f) | - | - | - | - | 63.4 | 52.3 | 77.0 | 65.5 | 85.4 |

**Long-sequence training.** Efficiently incorporating longer sequences during training (line d) helps the model to properly utilize its long-term memory, enabling generalization to indefinitely long sequences and leading to a notable +1.9 improvement in AssA.

Our final query propagation method (line d) improves MeMOTR's association accuracy by +5.2 (line f), and MOTR's by an impressive +17.3 (line e).

## 6 LIMITATIONS

Following the tracking-by-propagation paradigm, our model drops tracklets that are inactive for more than $N_{miss}$ frames to decrease the risk of ID switches. However, in some datasets like SportsMOT (Cui et al., 2023) football players may disappear from the camera view for multiple seconds, outliving the $N_{miss}$ threshold. We argue that future work should complement tracking-by-propagation with long-term re-identification to tackle this issue. Furthermore, in this paper, we introduced Samba, a set-of-sequences model. Our ablation study (Tab. 4) shows that Samba significantly outperforms the already strong SSM baseline. However, this comes with the trade-off of increased computational complexity. In particular, SSMs have linear complexity in time and linear complexity in the number of sequences (tracklets) independently modeled. Samba retains linear-time complexity, which enables it to track for indefinitely long-time horizons, but quadratic complexity in the number of sequences due to the use of self-attention in memory synchronization. Our ablations show that this trade-off is worth the performance improvement.

## 7 CONCLUSION

The proposed SambaMOTR fully leverages the sequential nature of the tracking task by using our set-of-sequences model, Samba, as a query propagation module to jointly model the temporal history of each tracklet and their interactions. The resulting tracker runs with linear-time complexity and can track objects across indefinitely long sequences. SambaMOTR surpasses the state-of-the-art on all benchmarks, reporting significant improvements in association accuracy compared to prior work.

ACKNOWLEDGMENTS

This work was supported in part by the Max Plank ETH Center for Learning Systems.

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
