APPENDIX

In this appendix, we report additional discussions and experiments. First, we provide background on sequence models in App. A. Then, we report additional implementation details for SambaMOTR in App. B. We show a schematic illustration of the Samba block in Fig. A and our method components in Fig. B. Finally, we provide additional results App. C, conducting several ablation studies on specific design choices that contributed to SambaMOTR's performance.

## A    BACKGROUND ON SEQUENCE MODELS.

**Sequence Models.** Sequence models are a class of machine learning models dealing with sequential data, *i.e.* where the order of elements is important. Applications of sequence models are widespread across different fields, such as natural language processing (Vaswani et al., 2017; Gu & Dao, 2023), time series forecasting (Wen et al., 2022) and video analysis (Venugopalan et al., 2015). Several architectures have been proposed to process sequences, each with its own strengths and limitations. *Recurrent neural networks (RNNs)* handles sequential data by maintaining a hidden state that updates as the network processes each element in a sequence. However, RNNs often struggle with long sequences due to issues like vanishing or exploding gradients (Pascanu et al., 2013). *Long short-term memory (LSTM)* networks (Hochreiter & Schmidhuber, 1997) introduce gating units to mitigate RNN's vanishing gradient problem. *Transformers* (Vaswani et al., 2017) rely on self-attention mechanisms to weigh the importance of different parts of the input data. Unlike RNNs and LSTMs, transformers process entire sequences simultaneously, making them efficient at modeling long-range dependencies at the cost of quadratic computational complexity wrt. sequence length. Building on the idea of modeling temporal dynamics like RNNs and LSTMs, structured state-space models (Gu et al., 2021) introduce a principled approach to state management inspired by classical SSMs (Kalman, 1960). Despite excelling at modeling long-range dependencies in continuous signals, structured SSMs lag behind transformers on discrete modalities such as text. Recently, selective state-space models (Mamba) (Gu & Dao, 2023) improved over prior work by making the SSM parameters input-dependent, achieving the modeling power of Transformers while scaling linearly with sequence length.

**Set-of-sequences Models.** Only few approaches (Yang et al., 2017; Amiridi et al., 2022; Wu et al., 2024) explore the task of set-of-sequences modeling, which we define as the task of simultaneously modeling multiple temporal sequences and their interdependencies to capture complex relationships and interactions across different data streams. Set-of-sequences modeling has applications in multivariate time series analysis (Amiridi et al., 2022), dynamic graph modeling (Wu et al., 2024), and sensor data fusion (Yang et al., 2017). However, existing techniques involve complex and expensive designs. We here introduce Samba, a linear-time set-of-sequences model based on the synchronization of multiple selective state-space models to account for the interaction across sequences.

## B    SAMBAMOTR - ADDITIONAL DETAILS

SambaMOTR builds on Samba to introduce linear-time sequence modeling in tracking-by-propagation, treating each tracklet as a sequence of queries and autoregressively predicting the future track query. By inducing synchronization on the SSMs' memories across an arbitrary number of sequences, Samba elegantly models tracklet interaction and query propagation under occlusions.

### B.1    SAMBA

We illustrate a Samba set-of-sequences model in Fig. A. A Samba model (Fig. A) is composed of a set of siamese Samba units (one for each sequence being modeled) with shared weights. Each Samba unit is synchronized with others through our synchronized SSM layer. In particular, a Samba unit is composed of $N$ non-linear Samba blocks. To obtain a non-linear Samba unitblock that can be embedded into a neural network, we wrap the synchronized SSM layer following the Mamba (Gu & Dao, 2023) architecture. A linear projection expands the input dimension $D$ by an expansion factor $E$, followed by a causal convolution and a SiLU (Hendrycks & Gimpel, 2016) activation before being fed to the sync SSM layer. The output of a residual connection is passed to a SiLU before being multiplied by the output of the synchronized SSM and passed to an output linear projection. More-

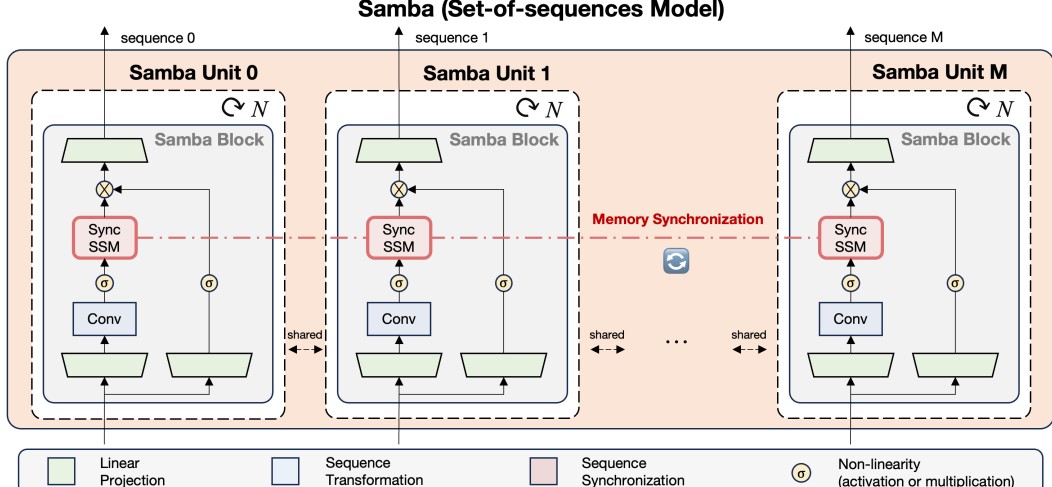

Figure A: **Illustration of our Set-of-sequences Model block.** Our set-of-sequences model Samba simultaneously processes an arbitrary number $M$ of input sequences. Each sequence is processed by a Samba unit, synchronized with the others thanks to our synchronized state-space model. All Samba units share weights and are composed of a stack of $N$ Samba blocks. A Samba block has the same architecture as a Mamba block, but it adopts our synchronized SSM to synchronize long-term memory representations across the individual state-space models.

over, we replace Mamba's RMSNorm (Zhang & Sennrich, 2019) with LayerNorm (Ba et al., 2016) for consistency with the detector. Finally, we repeat $N$ such Samba blocks, interleaved with standard normalization and residual connections, to form a Samba unit. The resulting set-of-sequences model is linear-time, supports a variable number of sequences that start and end at different time steps, and models long-range relationships and interdependencies across multiple sequences.

## B.2 SCHEMATIC ILLUSTRATION OF OUR CONTRIBUTIONS

We provide a schematic illustration of our contributions towards building Samba in Fig. B, disentangling them from one another to make the functioning of each component clear.

Mamba is the underlying sequence model, shown in the first row (Mamba). The second row depicts our strategy to deal with uncertain observations by ignoring them in the state update (Occlusion Masking). Synchronization across multiple sequence models using our synchronization module to let their hidden states communicate to model sequence interaction is shown in the third row (Sync). The last row illustrates our efficient training strategy to learn long-range dynamics from longer sequences at a comparable computational expense for backpropagation (Longer).

Each of these components is ablated in Tab. 4 by incrementally adding them within our framework, showing the effectiveness of each towards the impressive final performance of SambaMOTR.

## C ADDITIONAL RESULTS

We report additional results on the popular MOT17 pedestrian tracking benchmark in App. C.1. We extend our ablation study in App. C.2, investigating the effectiveness of synchronization, the use of positional embeddings and the effectiveness of residual prediction.

### C.1 MOT17

While MOT17 served as a benchmark of paramount importance to advance the state of current multiple object tracking algorithms, its very small size is reducing its significance as a training dataset. Since MOT17 only counts 7 training videos, modern tracking solutions complement its training with additional detection datasets and increasingly stronger detectors to improve the overall

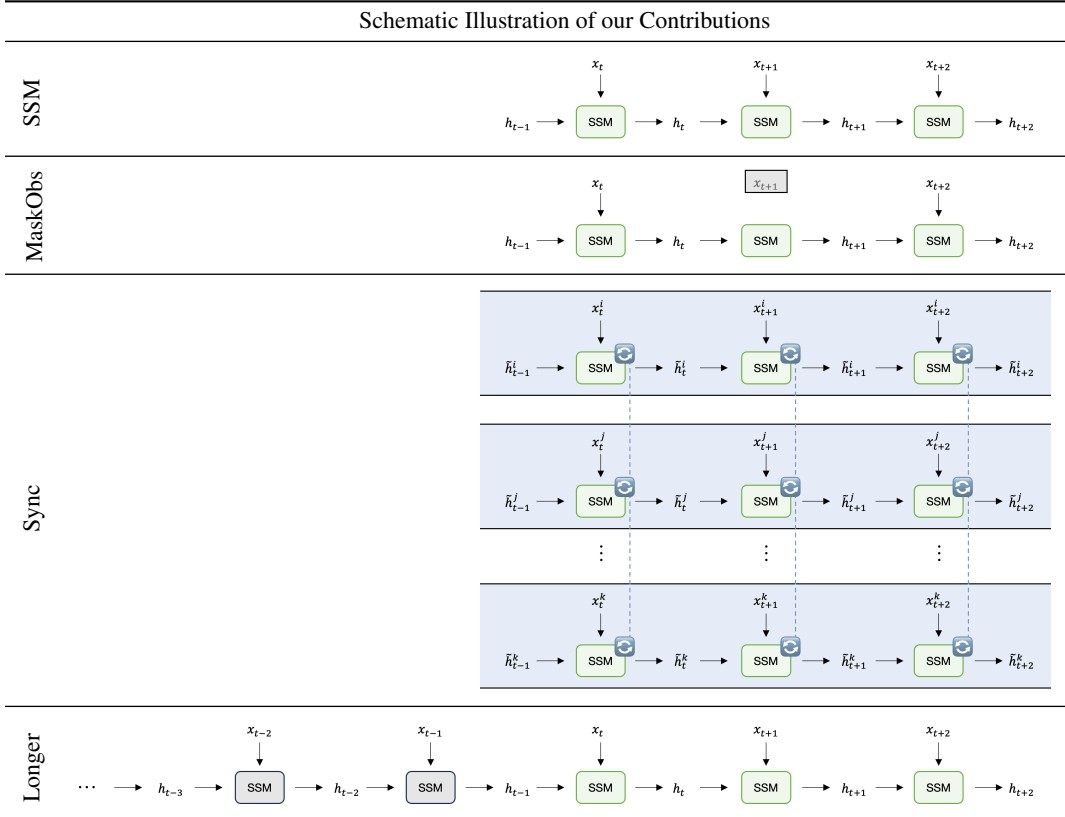

Figure B: Schematic illustration of our contributions (as ablated in Tab. 4). State-space model (SSM) blocks at timesteps with gradient applied are in green, and blocks without gradient are in grey.

tracking performance and top the leaderboard. However, such expedients are deviating from the study of fundamental tracking solutions and focusing more on engineering tricks. Moreover, due to its highly linear motion, its small size (only 7 videos), and the subsequent need for training on additional detection datasets, end-to-end tracking methods do not provide additional advantages over more naive Kalman-filter-based methods. For this reason, we preferred to it other more modern and meaningful datasets in the main paper, *i.e.* DanceTrack (Sun et al., 2022), SportsMOT (Cui et al., 2023) and BFT (Zheng et al., 2024), which allows us to study the importance of modeling tracklets interaction and of implicitly learning motion and appearance models to cope with the underlying non-linear motion, appearance and pose changes of the objects. Nevertheless, we here compare with the state-of-the-art for completeness and show comparable performance to previous tracking-by-propagation methods.

## C.2   ABLATION STUDIES

We here complement the ablation study in Sec. 5.4 with additional experiments on specific SambaMOTR's design choices. All ablations are based on the final version of our method, including all contributions as in Tab. 4 line d.

**Ablation on different formulations of synchronization.** In Tab. C, we ablate on different formulations of state synchronization and report the corresponding state update equation for each option. In particular, the first row (Sync: -) does not apply state synchronization and is equivalent to using Mamba as a query propagation module together with our occlusion masking and efficient longer training strategy as explained in Sec. 4.3. Since this option does not model tracklet interaction, it reports the lowest performance. We then compare synchronizing the hidden state before (prior) or after (posterior) and find that synchronizing the posterior is more effective. We attribute this to the

Table A: State-of-the-art comparison on MOT17 (Milan et al., 2016). Best tracking-by-propagation method in **bold**; best overall underlined. For a fair comparison with the only MeMOTR's published result, we also adopt DAB-Deformable-DETR.

| Methods | Detector | HOTA | AssA | DetA | IDF1 | MOTA |
|---|---|---|---|---|---|---|
| *Tracking-by-detection:* | | | | | | |
| CenterTrack (Zhou et al., 2020) | CenterNet (Duan et al., 2019) | 52.2 | 51.0 | 53.8 | 64.7 | 67.8 |
| FairMOT (Zhang et al., 2021) | | 59.3 | 58.0 | 60.9 | 72.3 | 73.7 |
| TransTrack (Sun et al., 2020) | | 54.1 | 47.9 | 61.6 | 63.9 | 74.5 |
| TransCenter (Xu et al., 2022) | Deformable DETR (Zhu et al., 2020) | 54.5 | 49.7 | 60.1 | 62.2 | 73.2 |
| MeMOT (Cai et al., 2022) | | 56.9 | 55.2 | - | 69.0 | 72.5 |
| GTR (Zhou et al., 2022) | CenterNet2 (Zhou et al., 2021) | 59.1 | 57.0 | 61.6 | 71.5 | 75.3 |
| DeepSORT (Wojke et al., 2017) | | 61.2 | 59.7 | 63.1 | 74.5 | 78.0 |
| SORT (Bewley et al., 2016) | | 63.0 | 62.2 | 64.2 | 78.2 | 80.1 |
| ByteTrack (Zhang et al., 2022) | | 63.1 | 62.0 | 64.5 | 77.3 | 80.3 |
| OC-SORT (Cao et al., 2023) | YOLOX-X (Ge et al., 2021) | 63.2 | 63.4 | 63.2 | 77.5 | 78.0 |
| QDTrack (Pang et al., 2021) | | 63.5 | 62.6 | 64.5 | 77.5 | 78.7 |
| C-BIoU (Yang et al., 2023) | | 64.1 | 63.7 | 64.8 | 79.7 | 81.1 |
| MotionTrack (Qin et al., 2023) | | 65.1 | 65.1 | 65.4 | 80.1 | 81.1 |
| *Tracking-by-propagation:* | | | | | | |
| TrackFormer (Meinhardt et al., 2022) | Deformable DETR (Zhu et al., 2020) | - | - | - | 68.0 | 74.1 |
| MOTR (Zeng et al., 2022) | | 57.2 | 55.8 | 58.9 | 68.4 | 71.9 |
| MeMOTR (Gao & Wang, 2023) | DAB-Deformable DETR (Liu et al., 2022) | 58.8 | 58.4 | 59.6 | 71.5 | 72.8 |
| SambaMOTR (ours) | | 58.8 | 58.2 | 59.7 | 71.0 | 72.9 |

opportunity to compensate for occluded observations in the current frame with the dynamics from other unoccluded tracklets to better model track query propagation through occlusions.

**Ablation on the effect of synchronization on hard DanceTrack sequences.** In Tab. B, we report the performance on hard sequences of the DanceTrack test set for two SambaMOTR with and without synchronization. We select the top-6 hardest sequences for the version without synchronization and show that utilizing synchronization greatly improves the overall metrics.

**Ablation on the use of query positional embeddings in Samba.** In Tab. D, we ablate on the addition of positional embeddings to the track embeddings before feeding them to Samba. We find that positional embeddings are very beneficial to Samba, arguably because they enable to implicitly learn non-linear motion models.

**Ablation on the prediction of residual vs. full queries with Samba.** In Tab. E, we ablate on the output format of our Samba-based query propagation module. We compare two versions: one that directly outputs the final track queries with Samba, and one that predicts a residual over the track queries used to detect in the current frame. We find that learning a residual is significantly more effective than directly predicting the final track query.

**Ablation on the query propagation strategy through occlusions.** We compare two query propagation strategies for occluded track queries in Tab. F. First, we evaluate our model with MeMOTR's (Gao & Wang, 2023) query propagation strategy (Freeze), which freezes the last observed state - *i.e.* the last track query that generated a confident detection - and memory until the tracklet is detected again in a new frame. Next, we compare this with actively propagating occluded track queries and their memory through occlusions using our MaskObs strategy (Sec. 4.3). We find that MaskObs outperforms Freeze: by inferring a tracklet's future state during occlusions using only its past memory and interactions with other observed objects, it keeps tracklets alive longer.

Table B: **Ablation on the effect of synchronization** on difficult sequences on DanceTrack test.

| Sequence | Synchronization | | | | | |
| --- | --- | --- | --- | --- | --- | --- |
| | ✗ | | | ✓ | | |
| | HOTA | DetA | AssA | HOTA | DetA | AssA |
| dancetrack0046 | 34.5 | 54.0 | 22.1 | 39.8 | 60.7 | 26.2 |
| dancetrack0085 | 39.6 | 60.9 | 25.9 | 40.3 | 63.9 | 25.5 |
| dancetrack0050 | 41.9 | 66.4 | 26.5 | 42.4 | 69.8 | 25.8 |
| dancetrack0036 | 42.9 | 74.7 | 24.7 | 48.4 | 77.5 | 30.3 |
| dancetrack0028 | 43.1 | 71.9 | 25.8 | 47.8 | 73.6 | 31.1 |
| dancetrack0009 | 43.4 | 73.6 | 25.7 | 48.0 | 75.2 | 30.7 |
| *average* | 40.9 | 66.9 | 25.1 | **44.5** | **70.1** | **28.3** |

Table C: **Ablation on memory synchronization positioning.** We report the state equation and performance on the DanceTrack test set for: (i) the baseline without synchronization (-); (ii) synchronization on the updated state prior to input contribution (Prior); (iii) synchronization on the fully-updated state (Posterior).

| Sync | State Equation | HOTA | AssA | DetA | IDF1 | MOTA |
| --- | --- | --- | --- | --- | --- | --- |
| - | $h_t^i = \bar{\mathbf{A}}^{\mathbf{i}}(t)h_{t-1}^i + \bar{\mathbf{B}}^{\mathbf{i}}(t)x_t^i$ | 66.0 | 56.4 | 77.5 | 69.5 | 86.7 |
| Prior | $h_t^i = \Gamma_{i \in \mathcal{T}}\left(\bar{\mathbf{A}}^{\mathbf{i}}(t)h_{t-1}^i\right) + \bar{\mathbf{B}}^{\mathbf{i}}(t)x_t^i$ | 66.1 | 56.7 | 77.3 | 70.0 | 86.4 |
| Posterior | $h_t^i = \Gamma_{i \in \mathcal{T}}\left(\bar{\mathbf{A}}^{\mathbf{i}}(t)h_{t-1}^i + \bar{\mathbf{B}}^{\mathbf{i}}(t)x_t^i\right)$ | **67.2** | **57.5** | **78.8** | **70.5** | **88.1** |

Table D: **Ablation on the use of positional embeddings.** We ablate on the addition of positional embeddings to the observed queries fed as input to the Samba module.

| Query Position | HOTA | AssA | DetA | IDF1 | MOTA |
| --- | --- | --- | --- | --- | --- |
| - | 65.6 | 56.2 | 76.7 | 69.3 | 85.4 |
| ✓ | **67.2** | **57.5** | **78.8** | **70.5** | **88.1** |

Table E: **Ablation on the use of residual prediction.** We evaluate two formats for the output of SambaMOTR's Samba module, *i.e.* direct query prediction (-) and prediction of a residual wrt. the track query from the previous frame (✓).

| Residual | HOTA | AssA | DetA | IDF1 | MOTA |
| --- | --- | --- | --- | --- | --- |
| - | 64.2 | 54.0 | 76.7 | 67.0 | 84.5 |
| ✓ | **67.2** | **57.5** | **78.8** | **70.5** | **88.1** |

Table F: **Ablation on strategies for tracking through occlusions.** We evaluate two strategies for tracking objects through occlusions: freezing the last observed track state (Freeze) as in MeMOTR, and propagating queries through occlusions using our MaskObs strategy.

| Strategy | HOTA | AssA | DetA | IDF1 | MOTA |
| --- | --- | --- | --- | --- | --- |
| Freeze | 65.9 | 56.6 | 76.8 | 69.7 | 86.3 |
| MaskObs | **67.2** | **57.5** | **78.8** | **70.5** | **88.1** |