# OpenReview forum: "Samba: Synchronized Set-of-Sequences Modeling for Multiple Object Tracking"
_ICLR.cc/2025/Conference — ICLR 2025 Spotlight_

### Official Review · Reviewer_WHQW · 2024-10-31

**Soundness:** 4
**Presentation:** 4
**Contribution:** 3
**Rating:** 8
**Confidence:** 4

**Summary:**

The paper introduces a tracking-by-propagation framework, SambaMOTR, which wanna tackles the question of how to model long-range dependencies within tracklets, interdependencies among tracklets, and the associated temporal occlusions. The core function of the  SambaMOTR is the Samba module, a novel linear-time set-of-sequences model designed to jointly process multiple tracklets by synchronizing the multiple selective state-spaces used to model each tracklet.  SambaMOTR is evaluated on DanceTrack, BFT, and SportsMOT datasets and achieves the state-of-the-art performance.

**Strengths:**

1. SambaMOTR introduces the novel linear-time set-of-sequences model designed to jointly process multiple tracklets by synchronizing the multiple selective state-spaces used to model each tracklet.
2. SambaMOTR achieves good tracking results on DanceTrack, BFT, and SportsMOT datasets.

**Weaknesses:**

1. The authors are advised to analyze the computational complexity.
2. The authors are advised to analyze why Samba is suitable for trajectory modeling and compare it with xLSTM, other SSMs, and RRNs.
3. The authors are advised to provide some tracking cases of SambaMOTR to demonstrate its superiority in trajectory modeling.

**Questions:**

see Weaknesses.

---

> ### Author Response · Authors · 2024-11-21
> **Authors Response**
>
> We thank the reviewer for recognizing the novelty and strong results of our approach, as well as for the high ratings on the soundness and presentation of our manuscript. The reviewer’s comments inspired the paragraph on **Efficiency Analysis and Comparison with Other Sequence Models** in the **General Comment** to this rebuttal, which we will also include in the camera-ready version. We here address the weaknesses raised in the review.
>
> ---
>
> **Computational Complexity**
>
> As detailed in the General Comment (see Efficiency Analysis), we have analyzed the computational complexity of our approach by evaluating the following metrics:
> 1. Peak memory consumption during training,
> 2. Throughput and latency during inference, and
> 3. The number of model parameters.
>
> This analysis compares SambaMOTR with the baseline MeMOTR and includes the integration of different sequence models within the Samba framework, as suggested by the reviewer. For detailed results and insights, please refer to the General Comment.
>
> ---
>
> **Comparison with Other Sequence Models**
>
> We thank the reviewer for emphasizing the importance of comparing Samba with alternative sequence models. Since Samba is inherently flexible regarding the underlying sequence model, we included comparisons with a representative state-space model (S4) and a state-of-the-art RNN-based approach (xLSTM).
>
> As reported in the General Comment (see Performance Comparison of Samba with Other Sequence Models), our analysis shows that Mamba-powered Samba outperforms both xLSTM and S4 in terms of tracking performance (HOTA). Additionally, Mamba is more parameter-efficient, requiring significantly fewer parameters (42.0M vs. 49.4M for xLSTM). These results further highlight the effectiveness and efficiency of Mamba as the sequence model within the Samba framework.

---

> > ### Comment · Reviewer_WHQW · 2024-11-25
> >
> > Thank you for your detailed response and for addressing the concerns raised in the review. The additional analysis on computational complexity and the comparison with alternative sequence models strengthen the manuscript. I would like to increase my score to 8.

---

> > > ### Author Response · Authors · 2024-11-26
> > >
> > > We are pleased that our rebuttal has addressed your concerns. Thank you for raising your score and supporting our submission.

---

### Official Review · Reviewer_Zen6 · 2024-11-03

**Soundness:** 3
**Presentation:** 3
**Contribution:** 2
**Rating:** 8
**Confidence:** 3

**Summary:**

The paper introduces the novel SAMBA model architecture for multi-object tracking. It takes a unique approach to tracking in a tracking-by-propagation framework, building on top of selective state-space models captured by Mamba models. Additional self-attention is used to model dependency between tracks. The method is evaluated on three tracking datasets, and it demonstrates a new state-of-the-art performance.

**Strengths:**

(1) The presented method achieves strong performance.

(2) the reason to integrate SSMs into a tracking-by-propagation framework is well motivated.

(3) The ablation studies conducted verified the introduced modifications.

(4) The paper is well written; the provided illustrative examples showcase strong results.

**Weaknesses:**

(1) There is a lack of qualitative comparison between prior works and the proposed method. It is helpful to see qualitative differences between methods' outputs to highlight what improvements in the models' output are contributing to the improvement in the performance.

(2) On L137, it is stated that  SambaMOTR has "the same GPU memory requirements". However, the paper lacks any measurements to back up this claim. Overall, there is emphasis placed on efficiency in the paper, so it would be good to include some measurements and comparisons to show these points. For example, including peak memory requirements and FLOPS would illustrate this point.

(3) While the method shows strong results, chiefly due to architectural modifications, it is worth asking whether efficiency gains enable larger (higher capacity) models, which lead to improvements or whether the SSMs formulation provides better biases for the tracking problem. While it might be difficult to disentangle such factors, the paper could help guide future research in this area by reporting some (albeit not great) proxies, such as learnable parameter counts, etc.

**Questions:**

(1) Would it be possible to include a comparison of SambaMOTR on top of YOLOX-X to have an apple-to-apples comparison with prior tracking-by-detections works?

---

> ### Author Response · Authors · 2024-11-21
> **Authors Response**
>
> We appreciate the reviewer’s positive feedback on the motivation of our work, its strong performance, the extensive ablation studies, and the clarity of our manuscript. We here address the remaining weaknesses and questions raised by the reviewer. Regarding qualitative comparisons to prior work, while the rebuttal format does not permit inclusion of such results, we commit to providing additional qualitative comparisons in the camera-ready version, specifically focusing on tracking under occlusion.
>
> ---
>
> **GPU Memory Requirements**
>
> We acknowledge that our claim in line 137 lacked supporting evidence, and we thank the reviewer for highlighting this. In particular, in line 137 we claimed that our efficient training recipe to learn from longer sequences allowed "improving the tracking performance while maintaining the same GPU memory requirements as previous methods."
>
> As suggested, we have conducted an efficiency analysis of SambaMOTR and combined it with WHQW's suggestions to analyze Samba with different sequence models. Reported in the **General Comment - Efficiency Analysis**, this analysis includes metrics such as peak memory consumption, throughput, latency, and parameter count. Regarding GPU memory, our results show that SambaMOTR using Mamba without synchronization has the same peak memory consumption as MeMOTR. Introducing synchronization increases the peak memory by only 0.5 GB.
>
> More specifically, our claim in line 137 refers to the efficient training strategy we use for learning from longer sequences. Thus, in the following table, we report the peak memory consumption during training when training on 5 frames vs. when training on 10 frames with our strategy. This strategy allows training on longer sequences without increasing peak memory consumption. Moreover, it results in improved tracking performance, as evidenced by the HOTA scores.
>
> | **Method** | **Type** | **Sequence Model** | **Sync** | **Longer** | **Peak Memory (MiB)** | **HOTA** |
> |------------|----------|---------------------|----------|------------|-----------------------|----------|
> | Ours       | SSM      | Mamba               | ✓        | -          | 16.4                 | 65.9     |
> |            |          | Mamba               | ✓        | ✓          | 16.4                 | 67.2     |
>
> ---
>
> **Disentangling Performance Improvements from Model Capacity**
>
> We understand the reviewer’s concern regarding the source of our performance improvements. The reviewer mentioned that "it is worth asking whether efficiency gains enable higher capacity models, which lead to improvements or whether the SSMs formulation provides better biases for the tracking problem." To address this, we compared SambaMOTR’s parameter count to that of MeMOTR, considering different sequence models, state dimensions, and the presence or absence of synchronization.
>
> As detailed in the **General Comment - Efficiency Analysis**, our findings indicate the following:
> 1. Mamba introduces fewer parameters than the heuristics-based propagation head of MeMOTR while achieving better performance.
> 2. The additional parameters introduced by synchronization are proportional to the hidden state dimension. For example, using a hidden state with four channels results in a dimensionality of 1024x4=4096 for the synchronization module’s feedforward network. This configuration increases the parameter count from 42.0M (baseline without synchronization) to 54.6M. Keeping the state dimension to 1 significantly lowers the parameter cost for synchronization, increasing the total parameter count to only 43.6M.
>
> Crucially, even without synchronization, our baseline model with 42.0M parameters outperforms MeMOTR. Thus, the addition of synchronization and the choice of hidden state size can be adapted to match the practitioner's computational requirements.
>
> As the reviewer already mentioned, we hope that this analysis provides valuable guidance for future research.
>
> ---
>
> **Comparison of SambaMOTR on Top of YOLO-X**
>
> While we agree with the reviewer that Samba can theoretically be extended to the tracking-by-detection paradigm by replacing the Kalman filter to jointly model motion and appearance trajectories, we believe that this analysis is better suited for a separate study. We are actively working on this extension.
>
> Since the current paper targets end-to-end tracking-by-propagation, we maintain that the most meaningful comparison is between MeMOTR and SambaMOTR.

---

> > ### Comment · Reviewer_Zen6 · 2024-11-24
> >
> > I thank the authors for the added details. I do not have further questions. I will keep my original rating

---

> > > ### Author Response · Authors · 2024-11-26
> > >
> > > We are pleased that the additional details we provided have clarified your questions. We sincerely thank you for the meaningful feedback and for appreciating our work.

---

### Official Review · Reviewer_dEH5 · 2024-11-04

**Soundness:** 2
**Presentation:** 2
**Contribution:** 3
**Rating:** 6
**Confidence:** 5

**Summary:**

The paper introduces SambaMOTR for multiple object tracking in videos. This method aims to improve tracking by utilizing temporal information about object movements to model historical object data, and occlusions, thereby modeling long-range dependencies, implicit motion, and appearance between tracklets.

**Strengths:**

- Overall, the experiments are extensive with different alterations in modeling provided in the main paper and appendix.

**Weaknesses:**

- Visual comparisons show that the Samba autoregressive model performs better than memory/motion models should be provided to substantiate the claims.
- The interpretations of system $\mathbf{A}$ and control $\mathbf{B}$ matrices in the context of object tracking are unclear. Providing clearer definitions of notations would be helpful.
- I understand the intention to separate two paradigms tracking-by-detection (TBD) and tracking-by-propagation (TBP), but the difference is subtle as the TBP still needs bounding boxes to initiate the states, and needs a detector to recognize newly appearing objects. Then, propagation alone is simply not practical in this problem of multiple object tracking.
- Then it should be fair to compare with TBD methods, and the performance still falls short behind classic motion models as reported in Table A.
- While I value the effort in development, the innovation in the method appears to be limited, as it essentially represents a straightforward combination of MOTR and Mamba and the motivation is not really compelling.

**Questions:**

- Just to confirm, does the state space model operate on the coordinate domain (bounding boxes) or the visual domain (pixel values)? If it does operate on the coordinate domain, how can the auto-regressed output boxes be refined to fit the subjects without looking at visual features?

---

> ### Author Response · Authors · 2024-11-21
> **Authors Response (Part 1: Questions)**
>
> We thank the reviewer for praising the extensive experiments and for providing constructive criticism. We here address the **questions** raised in the review.
>
> ---
>
> **Does the state space model operate on the coordinate domain (bounding boxes) or the visual domain (pixel values)?**
>
> The reviewer raises an important point: if Samba operated solely on the coordinate domain, auto-regressed output boxes could not be refined to accurately fit subjects without visual feature input.
>
> Our Samba module operates on both the coordinate and visual domains. As explained in **Sec. 4.3 - Query Propagation with Samba**, we construct the inputs to the sequence model by combining the bounding box coordinates and query embeddings outputted by the detector’s transformer decoder. Specifically, we sum the positional embedding of the predicted bounding box coordinates with their corresponding visual representation (query embedding).
>
> To validate this design, Table D (in the Appendix) compares the effects of including or excluding the positional embedding. Results show that combining coordinate and visual representations consistently outperforms using visual features alone. This illustrates how our propagation module can simultaneously model object trajectories in both coordinate and visual spaces.
>
> We hope that this answers the reviewer's question, potentially clarifying that one of the novelties also lies in how our propagation module can simultaneously model object trajectories both in the coordinate and visual space.

---

> ### Author Response · Authors · 2024-11-21
> **Authors Response (Part 2: Weaknesses)**
>
> We here address the **weaknesses**.
>
> ---
>
> **Interpretation of System A and Control B Matrices**
>
> We agree that a clearer explanation of the system $A$ and control $B$ matrices in the context of object tracking will benefit readers. Using the notation from Eqs. 3a-3c:
> - For a tracklet $i$, the observation $x_t^i$ at time $t$ is a joint representation of the object’s current location and visual appearance.
> - The hidden state $h_{t-1}^i$ represents the object’s history, including its past motion, appearance, and pose changes.
>
> In the memory update equation (Eq. 3a):
> $$
> h_t^i = \mathbf{\bar{A}}^i(t)h_{t-1}^i + \mathbf{\bar{B}}^i(t)x_t^i,
> $$
> we can disentangle it into:
> 1. $\mathbf{\bar{A}}^i(t)h_{t-1}^i$: Represents system dynamics, predicting the expected future hidden state (where the object will move and how it will look like) based only on past motion and appearance.
> 2. $\mathbf{\bar{B}}^i(t)x_t^i$: Incorporates the current observation to correct the predicted hidden state.
>
> Thus, $\mathbf{\bar{A}}^i(t)$ represents the tracklet's temporal dynamics, while $\mathbf{\bar{B}}^i(t)$ refines these dynamics using the latest detection. This dual mechanism ensures robust modeling of both long-term dependencies and immediate corrections. We will include this clarification in the camera-ready version.
>
> ---
>
> **Tracking-by-Detection (TbD) vs. Tracking-by-Propagation (TbP)**
>
> We argue that the distinction between TbD and TbP is significant, both conceptually and in practice. To illustrate:
> - In **TbD**, the detector operates independently for each frame: $y_t = f(I_t)$, where $f()$ is the detection function. Detection results $y_t$ are post-processed with a downstream association step.
> - In **TbP**, the detector is conditioned on previous detections: $y_t = f(I_t, g(y_{t-1}))$, where $g()$ is the propagation module that propagates prior outputs. This conditioning is done by propagating the output embeddings corresponding to detected objects at time $t-1$ and appending them ($g(y_{t-1})$) at time $t$ to the list of detection queries before feeding them to the transformer decoder at time $t$. This conditioning fundamentally changes how the detector operates during both training and inference, creating a distinct paradigm. In particular, TbP methods can rely on end-to-end training over videos, bridging the training-inference gap and leveraging temporal dependencies.
>
> We will improve the clarity of this distinction in the paper.
>
> ---
>
> **Comparison with Tracking-by-Detection (TbD)**
>
> Based on the assumption that TbD and TbP are not inherently different, the reviewer points out:
> > "Then it should be fair to compare with TBD methods, and the performance still falls short behind classic motion models as reported in Table A."
>
> While we hope to have clarified in the previous section that the difference between the two families of approaches is significant, we acknowledge the reviewer’s observation that TbP methods fall short on MOT17 compared to TbD (Table A).
>
> However, this limitation stems from the training requirements of TbP, not an inherent inferiority of the approach.
>
> TbP methods rely on end-to-end training over videos, bridging the training-inference gap and leveraging temporal dependencies. In contrast, TbD methods use independent frames for training, allowing them to benefit from large-scale datasets like CrowdHuman. Since MOT17 is a small dataset with only 7 annotated videos (line 897), the scarcity of video information limits TbP training effectiveness.
>
> To mitigate this, we augmented the MOT17 training set with pseudo-videos generated by applying multiple data augmentations on individual CrowdHuman images. While this improves results, pseudo-videos cannot fully replicate real-world dynamics, leading to performance gaps due to the mismatch between the end-to-end training and the autoregressive inference. This limitation highlights an important area for future research, and we hope the reviewer appreciates our transparency in acknowledging it (Sec. C.1), as recommended by ICLR guidelines.
>
> ---
>
> **Motivation and Novelty**
>
> While the reviewer questions the motivation and novelty of our approach, we believe these aspects are strongly supported by our contributions and results:
> 1. **Motivation:** Current end-to-end tracking methods fail to model long-range dependencies and object interactions effectively. Samba addresses this gap by introducing learnable set-of-sequences models in TbP, as appreciated by other reviewers and confirmed by our performance improvements.
> 2. **Novelty:** Samba is not a mere combination of MOTR and Mamba. Samba is flexible to other sequence models (see General Comment). Additional key contributions of our submission (see ablations) include synchronization, MaskObs, and efficient training on longer sequences. These components are indispensable to our framework and collectively enable its strong performance.
>
> Hoping this rebuttal addresses the concerns, we remain available for discussion.

---

> > ### Comment · Reviewer_dEH5 · 2024-11-26
> >
> > Thank you for your rebuttal. Despite the provided explanation, I maintain several critical observations about the claims:
> >
> > 1. Advantages of TbP methods: Because "TbP methods rely on end-to-end training over videos, bridging the training-inference gap and leveraging temporal dependencies" as in your statements, I would expect the temporal reasoning and performance are better than discrete models like classical motion models compared to TbD. Additionally, "efficient training on longer sequences" would support my expectations to give superior performance.
> >
> > 2. Tracking-by-Detection (TbD) vs. Tracking-by-Propagation (TbP): For TbP, do you still need an initiated state to be propagated? Do you need a detector to perform periodically so that the framework can get newly detected objects? If it is yes, then I see the TbP is just a special case of TbD.
> >
> > Overall, I recommend the authors revise their language to present a more measured and evidence-based assessment that reflects the actual performance improvements over discrete models.

---

> > > ### Author Response · Authors · 2024-11-26
> > > **Authors Response (Part 3 - Remaining Concerns)**
> > >
> > > We thank the reviewer for their reply, and we are glad that our previous comments resolved some of their concerns. We will now address the remaining ones.
> > >
> > > ---
> > >
> > > **Advantages of TbP Methods**
> > >
> > > Since the reviewer suggests an evidence-based method to justify the superiority of TbP to TbD, we have structured this rebuttal to focus on empirical evidence. As the reviewer mentioned:
> > > > "TbP methods rely on end-to-end training over videos, bridging the training-inference gap and leveraging temporal dependencies. As in your statements, I would expect the temporal reasoning and performance to be better than discrete models like classical motion models compared to TbD. Additionally, 'efficient training on longer sequences' would support my expectations to give superior performance."
> > >
> > > As we have shown in the paper and backed up by empirical evidence (Table 1, Table 2, and Table 3), TbP indeed provides far superior performance to TbD. We want to stress that we are not introducing the first TbP method ourselves but rather defined a taxonomy in the paper for ease of comparison for future research. Nevertheless, for each analyzed dataset, we provide a summary of the best TbD method vs. our TbP method (SambaMOTR). This should convince the reviewer that, when video training data is available, TbP is a superior paradigm  and SambaMOTR is the strongest method.
> > >
> > > | **Dataset**     | **Type** | **Method**            | **HOTA** | **AssA** | **DetA** |
> > > |------------------|----------|-----------------------|----------|----------|----------|
> > > | DanceTrack       | TbD      | C-BIoU (Yang et al., 2023) | 60.6     | 45.4     | 81.3     |
> > > |                  | TbP      | SambaMOTR            | 67.2     | 57.5     | 78.8     |
> > > | BFT              | TbD      | OC-SORT (Cao et al., 2023) | 66.8     | 68.7     | 65.4     |
> > > |                  | TbP      | SambaMOTR            | 69.6     | 73.6     | 66.0     |
> > > | SportsMOT        | TbD      | TransTrack (Sun et al., 2020) | 68.9     | 57.5     | 82.7     |
> > > |                  | TbP      | SambaMOTR            | 69.8     | 59.4     | 82.2     |
> > > | MOT17 (†)           | TbD      | MotionTrack (Qin et al., 2023) | 65.1     | 65.1     | 65.4     |
> > > |                  | TbP      | SambaMOTR            | 58.8     | 58.2     | 59.7     |
> > >
> > >
> > > As seen in the table, our method outperforms the best TbD competitors on datasets where sufficient video training data is available:
> > > - SambaMOTR outperforms C-BIoU by **+6.6 HOTA** and **+12.1 AssA** on DanceTrack.
> > > - SambaMOTR outperforms OC-SORT by **+2.8 HOTA** and **+4.9 AssA** on BFT.
> > > - SambaMOTR outperforms TransTrack by **+0.9 HOTA** and **+1.9 AssA** on SportsMOT.
> > >
> > > On MOT17 (†), however,  performance for TbP is worse because **abundant video data is not available**.  While TbD methods have historically relied on external image-level object detection datasets (e.g. CrowdHuman) to achieve state-of-the-art performance on MOT17, the lack of abundant video training data limits the effectiveness of TbP methods. As we mentioned in our previous comment (see **Authors Response (Part 2: Weaknesses) - Comparison with Tracking-by-Detection (TbD)**):
> > > - MOT17 consists of only 7 videos.
> > > - Our method and TbP methods, in general, are inherently designed for training on video sequences. Thus, training TbP methods on image-level datasets cannot leverage the properties of TbP.
> > >
> > > This highlights that:
> > > 1. MOT17 is not suitable for modern video-level training.
> > > 2. MOT17 is heavily reflective of object detection performance rather than tracking improvements.
> > >
> > > We have reported results on MOT17 for completeness and to guide the research community toward more suitable benchmarks for modern learning-based research, which we hope the reviewer and the community will appreciate.
> > >
> > > We hope this evidence-based reply convinces the reviewer of the superiority of SambaMOTR (and TbP) when video data is available.

---

> > > > ### Author Response · Authors · 2024-11-26
> > > > **Authors Response (Part 4 - Remaining Concerns)**
> > > >
> > > > **Tracking-by-Detection (TbD) vs. Tracking-by-Propagation (TbP)**
> > > >
> > > > > "For TbP, do you still need an initiated state to be propagated?"
> > > >
> > > > For TbP, the initial state is defined by the new object detections that are not associated with any existing track query.
> > > >
> > > > > "Do you need a detector to perform periodically so that the framework can get newly detected objects? If it is yes, then I see the TbP is just a special case of TbD."
> > > >
> > > > Yes, TbP needs an object detector in the loop. However, as we explained in the previous reply (see **Authors Response (Part 2: Weaknesses) - Tracking-by-Detection (TbD) vs. Tracking-by-Propagation (TbP)**), the object detector is used differently from TbD methods. In TbP, the object detector's transformer decoder is conditioned on the tracked objects ($y_t = f(I_t, g(y_{t-1}))$) so that they can be re-detected by the same propagated track query.
> > > >
> > > > In contrast, TbD methods apply the object detector independently for each frame ($y_t = f(I_t)$), and only then apply some association heuristics to associate the detected objects with old tracklets or use them to initialize new ones. Hence the name tracking-by-detection, which means that detection is done first, and association is a downstream task performed on the existing detections. This is also reflected by the fact that in TbD the object detector can be trained separately.
> > > >
> > > > In TbP, object detection and tracking are tightly coupled and jointly learned. The object detection stage is conditioned on the existing tracklets, and the association stage is unnecessary because the same propagated track query will automatically re-detect the same object instance.
> > > >
> > > > Thus, both mathematically and intuitively, TbP cannot be seen as a special case of TbD. We hope that the reviewer finds our reply convincing, and we will include this insightful discussion in the main paper to assist future readers and researchers.

---

> > > > > ### Author Response · Authors · 2024-11-26
> > > > > **Part 5 - Additional clarification on the taxonomy**
> > > > >
> > > > > After carefully inspecting the reviewer’s comments, we have just realized that the confusion arising from the taxonomy introduced (i.e. TbD vs TbP) stems from a mutual misunderstanding. We apologize for having initially misunderstood what the reviewer meant.
> > > > >
> > > > > **To clarify:** both families of methods aim at solving the same task, namely MOT, and thus both require an object detector by definition of the task. Thus, TbD and TbP share the same set of assumptions, receiving a video as a sequence of images and detecting and tracking object instances through time.
> > > > >
> > > > > In particular, the following **definition of TbD** is commonly accepted in literature: “First, objects are detected in each frame of the sequence and second, the detections are matched to form complete trajectories.” (from Laura Leal-Taixé’s Doctoral Thesis, “Multiple object tracking with context awareness“, 2014).
> > > > >
> > > > > Please notice that **TbP breaks this definition**: by conditioning the object detection stage on the tracking results from the previous timesteps, it violates the common definition of TbD, according to which detection and association are disentangled and sequential. Hence, the **necessity for a different categorization for TbP**.
> > > > >
> > > > > Thus, we would like to clarify that our taxonomy (TbD vs TbP) is meant as a finer categorization of how these families of methods approach MOT during both training and inference, each coming with unique trade-offs:
> > > > > - **TbD:**
> > > > >   - **Inference:** “First, objects are detected in each frame of the sequence and second, the detections are matched to form complete trajectories.” (from Laura Leal-Taixé’s Doctoral Thesis, “Multiple object tracking with context awareness“, 2014).
> > > > >   - **Training:** ⁠⁠TbD learns object detection independently from each video frame, optionally treating instance representation learning as a separate stage in its pipeline.
> > > > >   - **Advantage:** By being independent of video-level training, it can leverage large external object detection datasets and perform well on small datasets with simple motion such as MOT17, as object detection dominates performance (Table A).
> > > > >   - **Limitation:** Suffers on more challenging datasets where exploiting video information during training could improve performance (Tables 1, 2, 3).
> > > > >
> > > > > - **TbP:**
> > > > >   - **Inference:** TbP differs from TbD, since it conditions the object detection stage on the tracking results from the previous timesteps. Thus, it violates the common definition of TbD, according to which detection and association are disentangled and sequential.
> > > > >   - **Training:** TbP bridges the training-inference gap by directly learning end-to-end from ordered videos during training by making the object detection conditional on the previous timesteps. Consequently, detection and propagation training are tightly coupled and both necessary (unlike TbD where the detector can be trained separately).
> > > > >   - **Advantage:** When trained on videos, it bridges the training-inference gap, leveraging video information to achieve superior tracking performance (Tables 1, 2, 3).
> > > > >   - **Limitation:** Requires video data for training, making it less effective when trained on image-level datasets due to its architecture’s dependency on temporal information.
> > > > >
> > > > > In the manuscript, the split is not meant to avoid comparison to TbD methods but to emphasize the practical differences of each approach. As highlighted in the previous reply, **TbP indeed outperforms TbD whenever video information is available during training.**
> > > > >
> > > > > We sincerely hope that this has addressed the misunderstanding originating from our taxonomy. We thank the reviewer for pointing out their concerns on the taxonomy and we will add a paragraph detailing these differences in the next revision.
> > > > >
> > > > > In light of the recent clarifications, does the reviewer have any remaining questions? We are happy to clarify any remaining concerns so that we can improve our manuscript.

---

> ### Comment · Reviewer_dEH5 · 2024-11-26
>
> Thank you for your efforts in clarification. I hope to see these details discussed in the revision. In the light of this, I have increased my rating.
>
> Considering my suggestion, more qualitative comparisons show that the Samba autoregressive model performs better than memory/motion models should be provided to visually substantiate the claims.

---

> > ### Author Response · Authors · 2024-11-26
> >
> > We are glad that you appreciated our efforts, we will discuss these details in the revised manuscript and provide the additional qualitative comparisons. Thank you!

---

### Author Response · Authors · 2024-11-21
**General Comment**

We thank the reviewers for their constructive feedback and for recognizing the strengths of our work, including its novelty and uniqueness [Zen6, WHQW], strong performance [Zen6, WHQW], extensive experiments [dEH5, Zen6], and clarity of presentation [Zen6, WHQW].

We appreciate the suggestions regarding efficiency analysis [Zen6, WHQW] and comparisons with other sequence models [WHQW]. In response, we have conducted a detailed efficiency analysis and provided the requested comparisons, which are included in this General Comment. Additionally, we commit to incorporating further qualitative comparisons in the camera-ready version. Reviewer-specific questions and concerns are addressed in the individual responses.

---

**Efficiency Analysis and Comparison with Other Sequence Models**

We thank Zen6 and WHQW for emphasizing the importance of conducting an efficiency analysis to validate our claims regarding the minimal computational overhead introduced by Samba. Moreover, we thank WHQW for recommending a comparison of SambaMOTR’s performance using different sequence models.

For clarity and conciseness, we combine the two studies. Specifically, we evaluated SambaMOTR with various sequence models, including SSM-based approaches (S4, Mamba) and the recent RNN-based xLSTM, against the baseline MeMOTR. All SambaMOTR models use our additional contributions, such as MaskObs and the efficient training recipe for longer sequences. We trained the models on DanceTrack and report their tracking performance (HOTA) on the DanceTrack test set. The evaluation also includes the following metrics:
- Peak memory consumption (GB) after 100 iterations of multi-GPU training without activation checkpointing.
- Model throughput (images per second) at inference time, measured on a single NVIDIA RTX 3090 for a randomly selected DanceTrack video (0054).
- Model latency (ms) at inference time, measured on a single NVIDIA RTX 3090 for the same video.
- Number of model parameters (millions, M).
- The tracking performance (HOTA).

| **Method** | **Type** | **Sequence Model** | **Sync** | **Peak Memory (GB)** | **Throughput (img/s)** | **Latency (ms)** | **Parameters (M)** | **HOTA** |
|------------|----------|---------------------|----------|-----------------------|------------------------|------------------|---------------------|----------|
| MeMOTR     | -        | -                   | -        | 15.9                 | 18.5                  | 54.0            | 42.9               | 63.4     |
| **SambaMOTR**   | RNN      | xLSTM               | -        | 17.0                 | 18.7                  | 53.4            | 49.5              | 65.2     |
|            | RNN  | xLSTM               | ✓        | 17.5                 | 18.6                  | 54.0            | 55.8               | 66.7     |
|            | SSM      | S4                  | -        | 15.9                 | 18.7                  | 53.3             | 42.4               | 64.7     |
|            |  SSM        | S4                  | ✓        | 16.1                 | 18.6                  | 53.4             | 43.8               | 65.5     |
|            |  SSM        | Mamba               | -        | 15.9                 | 18.6                  | 53.7            | 42.0              | 66.1     |
|            |  SSM        | Mamba               | ✓        | 16.4                 | 18.6                  | 54.1             | 43.6               | **67.2** |

Our results demonstrate that Samba’s long-range set-of-sequences model introduces only a negligible increase in peak memory consumption, throughput, latency, and parameter count compared to MeMOTR’s heuristics-based memory model. All sequence models use a hidden state dimension of 1; increasing the state dimension leads to a proportional increase in synchronization parameters. Among the sequence models, Mamba proves to be the most parameter-efficient while achieving the best performance. Even without synchronization, Mamba outperforms MeMOTR with fewer parameters.

Crucially, these findings confirm that the observed performance improvements arise from the principled design of our propagation module, which effectively leverages long-range trajectory information, rather than from increased model capacity.

---

### Meta-Review · Area_Chair_6kQs · 2024-12-19

**Metareview:**

Summary
This paper proposes SambaMOTR based on a linear-time set-of-sequences model for multiple object tracking. It processes multiple tracklets simultaneously by synchronizing multiple selective state spaces. The method leverages temporal information about object movements to model historical data and handle occlusions, enabling the modeling of long-range dependencies, implicit motion, and appearance relationships between tracklets. SambaMOTR achieves state-of-the-art performance on the DanceTrack, BFT, and SportsMOT datasets.

Strengths & weaknesses
All reviewers recognized the strengths of the paper are strong experimental results and innovative ideas, and the weaknesses are lacking some technique details and more experimental results (visual comparisons, efficiency evaluations). After rebuttal, the weaknesses are well addressed by the authors. The final ratings are 8,8,6 and all reviewers recommend accepting this paper.

The AC concurs with the reviewers and recommends accepting this paper. In addition, the authors are encouraged to incorporate the suggested technical details and efficiency evaluations to further enhance the work.

**Additional Comments On Reviewer Discussion:**

In the author-reviewer discussion phase, the authors provided more technique details, analysis, and evaluations on efficiency, which well addressed the concerns raised by the reviewers. The final ratings are 8,8,6 and all reviewers recommend accepting this paper.

In addition, two reviewers highlighted the need for additional technical details and efficiency evaluations. The authors are encouraged to incorporate the suggested technical details and efficiency evaluations to further enhance the work.

---

### Decision · Program_Chairs · 2025-01-22

Accept (Spotlight)